# Critical Roles of Tumor Extracellular Vesicles in the Microenvironment of Thoracic Cancers

**DOI:** 10.3390/ijms21176024

**Published:** 2020-08-21

**Authors:** Lyna Kara-Terki, Lucas Treps, Christophe Blanquart, Delphine Fradin

**Affiliations:** 1CRCINA, INSERM, Université de Nantes, F-44000 Nantes, France; lyna.kara-terki@etu.univ-nantes.fr (L.K.-T.); lucas.treps@kuleuven.vib.be (L.T.); christophe.blanquart@inserm.fr (C.B.); 2Laboratory of Angiogenesis and Vascular Metabolism, Department of Oncology and Leuven Cancer Institute (LKI), KU Leuven, VIB Center for Cancer Biology, 3000 Leuven, Belgium

**Keywords:** exosome, mesothelioma, lung cancer, miRNA

## Abstract

Extracellular vesicles (EVs), such as exosomes, are critical mediators of intercellular communication between tumor cells and other cells located in the microenvironment but also in more distant sites. Exosomes are small EVs that can carry a variety of molecules, such as lipids, proteins, and non-coding RNA, especially microRNAs (miRNAs). In thoracic cancers, including lung cancers and malignant pleural mesothelioma, EVs contribute to the immune-suppressive tumor microenvironment and to tumor growth and metastasis. In this review, we discuss the recent understanding of how exosomes behave in thoracic cancers and how and why they are promising liquid biomarkers for diagnosis, prognosis, and therapy, with a special focus on exosomal miRNAs.

## 1. Introduction

Thoracic cancers, including lung cancers and malignant pleural mesothelioma (MPM), represent a global health burden primarily due to the lack of effective treatment options for most patients. Lung cancers can be divided into two major subtypes based on their cell morphologies: small-cell lung cancer (SCLC) (~15%) and non-small cell lung cancer (NSCLC) (~85%), which includes squamous cell carcinoma, adenocarcinoma, and large cell carcinoma [1]. Long-term smoking is one of its leading causes, whereas exposure to asbestos fibers has been associated with MPM for a long time now [2,3]. MPM is a cancer formed mainly in the pleura, a membrane covering lungs and the inner side of the ribs. MPM is usually asymptomatic during a protracted period (20 to 50 years), until it has reached an advanced stage.

In the tumor microenvironment (TME), the communication between tumor and immune cells takes center stage in cancer development and progression. Tumor exosomes (TEXs) are important players in this communication, and notably have the capacity to alter immune response especially by inducing macrophage M2 polarization and promoting tumor progression in numerous cancers, including lung cancers [4]. TEXs also facilitate progression, invasion, angiogenesis, metastasis and drug resistance. Some studies suggest that it could be due to the transfer of microRNAs (miRNAs) from tumor to recipient cells since they are largely enriched into exosomes [5,6]. miRNAs are a subset of small non–coding RNAs (from 19 to 22 nucleotides) known to regulate gene expression at the post–transcriptional level through mRNA silencing or degradation. As such, miRNAs exert pivotal regulatory roles in cells by regulating more than 60% of the genes in humans [7]. Because exosomes are stable in various biological fluids and carry cargos that in part mimic contents of parent cells, they are also of potential interest as non-invasive and easily accessible cancer biomarkers.

This review will summarize the current knowledge about the thoracic cancers–derived exosomes and their role in the TME, with a special focus on the miRNA content and transfer.

## 2. Exosome Biogenesis and Secretion

Exosomes are nano-sized vesicles generated inside cells during the maturation of endosomes into multivesicular bodies (MVBs). This is a multistep process starting with the invagination of the plasma membrane leading to the formation of early endosomes (Figure 1). Next, many small vesicles called intraluminal vesicles (ILVs) are formed inside them by the inward invagination of their membranes, and such endosomes are called MVBs [8]. The endosomal sorting complex required for transport (ESCRT) machinery is involved in this step. This machinery is composed of four complexes (ESCRT-0, I, II, III) working sequentially, but some evidence showed that vesicles can also be formed without them, involving for example tetraspanins [9] and/or lipids [10,11]. Finally, MVBs either fuse with lysosomes in order to hydrolyze their content, or with the plasma membrane to release the ILVs, called now exosomes, into the extracellular space. Similarly, the exosome secretion from the parent cell involves multiple steps: trafficking of MVBs to the plasma membrane, docking, fusion, and release [12].

Once released by parent cells, exosomes can interact with neighboring or distant cell via at least three mechanisms [13]: (i) interaction between exosome transmembrane proteins and the signaling receptors of recipient cells, (ii) exosome fusion with the plasma membrane of recipient cell and release of their content into the cytosol, (iii) exosome internalization into the recipient cell by endocytosis. Interestingly, it has been shown that exosomal miRNAs released in the cytoplasm are functional and may induce function and phenotype changes in recipient cells [6,14].

## 3. Exosome Content

During their formation, specific constituents of the parent cell content are trapped into exosomes. A wide variety of molecules are thus found in exosomes: lipids, proteins, and nucleic acids. According to the present version of the exosome content databases (ExoCarta [15], Vesiclepedia [16] or EVpedia [17]), more than 9700 proteins, 1100 lipids, 3400 mRNAs and 2800 miRNAs have been identified in various exosomes or small EVs.

### 3.1. Lipids

The exosome membrane is enriched in cholesterol, sphingomyelin, ceramides, phosphatidylserine, phosphatidylinositol, and phosphatidic acid [18]. Their lipids’ compositions impact their membrane fluidity, allowing them to be stable in the extracellular space, but also impacting their own formation. Indeed, membrane curvature is strongly dependent on membrane lipids, themselves conditional on the enzymes available in the cell, such as the sphingomyelinase 2 (nSMase2) [11] or phospholipase D2 (PLD2) [19].

### 3.2. Proteins

Exosomes contain many proteins, and some of them are highly enriched, such as tetraspanins (CD63, CD81, CD9), heat shock proteins (HSP70, HSP90), MVBs formation proteins (TSG101, Alix), or others, and could therefore be used as markers for exosome characterization [20]. Interestingly, exosomes also carry immune proteins, such as those of major histocompatibility complex I and II (MHC I/II) or PD-L1 (programmed death ligand 1), and can be consequently involved in the modulation of the immune response against the tumor (see parts 4 and 5).

### 3.3. Nucleic Acids

Exosomes are enriched in RNAs, especially in small RNA such as miRNAs. Exosomal miRNAs are not randomly incorporated into them, pointing out the existence of a mechanism for active sorting of specific miRNAs into exosomes. The exact mechanisms by which this selection is made are not totally understood. However, four potential pathways have been proposed. The nSMase2 was the first molecule to be involved in the miRNAs sorting into exosomes. In 2013, Kosaka and collaborators [21] reported that the overexpression of the nSMase2 increased the miRNAs levels in HEK293-derived exosomes and, in contrast, a decrease of its expression was associated with a lower content of miRNAs in exosomes. Another sorting mechanism was described the same year by Villarroya-Beltri et al. [22]. They described the involvement of the sumoylated version of the RNA-binding protein hnRNPA2B1 in the miRNAs sorting into T cells-derived exosomes. This protein recognizes a specific motif GAGAG called “EXO-motif” in the miRNA 3’end sequence and allows a selective loading of these miRNAs into exosomes. Since this discovery, other RNA-binding proteins including SYNCRIP (synaptotagmin binding, cytoplasmic RNA interacting protein) [23] and YBX1 (Y box binding protein 1) [24] have been identified for their role in the miRNA sorting. The third mechanism has been described by Kopper-Lalic et al. in 2014 [25]. They found that 3’end uridylated miRNAs were enriched into B cell-derived exosomes, whereas 3’end adenylated endogenous miRNAs stayed in the cell. This observation shows that modifications of the miRNA 3’end portion are linked to their loading into the exosome. The fourth pathway involves the miRNA-induced silencing complex (miRISC)-related pathway: mature miRNAs can interact with assembly proteins to form the miRISC complex mainly composed of the mature miRNA, its target mRNA, GW182, and AGO2 (Argonaute2). The knockout of AGO2 decreases the sorting of some miRNAs such as miR-451, miR-150, and miR-142-3p in HEK293T cell-derived exosomes, suggesting that AGO2 could be involved in the miRNA sorting into exosomes [26]. As outlined later, these sorting mechanisms could be important to translate exosomes into therapeutic applications.

The international society for extracellular vesicles (ISEV) recommends the use of the generic term extracellular vesicle (EV) as “particles naturally released from the cell that are delimited by a lipid bilayer and cannot replicate” [20]. Exosomal vesicles are included in these EVs but need to be rigorously characterized since it is often difficult to distinguish them once they leave the cell. Moreover, during their isolation, they are frequently mixed with other vesicles including microvesicles and apoptotic bodies. The term exosomes has been extensively used in the past but tend to be progressively replaced by the more accurate term, small EVs. In our review, we will use either term according to the publication source, but notably, sometimes exosomes have not been investigated according to the MISEV guidelines, and are probably in fact a mix of EVs.

## 4. Lung Cancer-Derived Exosomes Foster Pro-Tumorigenic Macrophages

Tumor-associated macrophages (TAMs) have increasingly become recognized as an attractive target in thoracic cancers. A number of macrophage-centered approaches have been investigated, including strategies to limit their infiltration or exploit their antitumor functions [27,28]. TAMs represent the majority of the tumor stroma and maintain intricate interactions with malignant cells within the tumor microenvironment (TME), largely influencing the outcome of the cancer growth and metastasis [29]. A growing number of studies have reported the capacity of tumor cells to communicate with immune cells from the TME through exosomes release, thus leading to immune suppression and tumor evasion. In the case of TAMs, TEXs induce mainly the polarization of macrophages into the M2-type, but few studies have been conducted in thoracic cancers [30]. A recent study by Pritchard et al. showed that lung tumor cell-derived exosomes promote M2 macrophage polarization from M0 THP1 cells [4]. Interestingly they found that non-tumor cell-derived exosomes do not have this effect. Even if the authors have not identified yet the exosome cargo responsible for this polarization, their results confirmed those previously described by Hsu et al., who have for their part described an underlying mechanism [31]. They showed that hypoxia stimulates the loading of the miR-103a in lung cancer-derived EVs (Figure 2). After their internalization by the monocytes, miR-103a targets the tumor suppressor PTEN (phosphatase and tensin homolog) to decrease its expression, leading to the activation of the PI3K/AKT (phosphatidylinositol-4,5-bisphosphate 3-kinase/AKT) and STAT3 (signal transducer and activator of transcription 3) signaling pathways, both enhancing the expression of CD163^+^ and CD206^high^ but also promoting factors such as IL-10, CCL18, (chemokine (C-C motif) ligand 18) and VEGF-A (vascular endothelial growth factor A). In contrast, the inhibition of miR-103a by an anti-miRNA decreases monocyte polarization into the M2-type.

## 5. Immunosuppressive Function of Thoracic Tumor-Derived Exosomes

Apart from macrophages, other immune cells are also affected by lung cancers and MPM exosomes. In this sense, Huang et al. showed that lung cancer-derived exosomes containing EGFR (epidermal growth factor receptor) induce tolerogenic dendritic cells (DCs) and regulatory T lymphocytes (Treg) which could suppress the tumor antigen-specific CD8^+^ T cells to promote tumor growth (Table 1) [32]. They also found that tumor-derived exosomes are enriched in EGFR compared to exosomes from chronic lung inflammation. Concerning Treg again, Yin et al. reported that lung cancer cells deliver increased levels of miR-214 to CD4^+^ T cells through exosomes, allowing thus the decrease of PTEN levels and the expansion of CD4^+^ CD25^high^ Foxp3^+^ Treg in a mouse model [33]. Myeloid-derived suppressor cells (MDSCs) can also be modified by lung cancer-derived exosomes. Indeed, Chalmin et al. showed that exosomes containing HSP72 could promote the suppressive activity of MDSCs through the activation of STAT3 by IL-6 (interleukin 6) [34]. More recently, it has been shown that exosomes derived from lung cancer cells express PD-L1 and play a role in immune escape by reducing the T-cell activity [35]. Finally, Berchem et al. showed that following their uptake by NK (natural killer) cells, vesicles derived from hypoxic lung cancer release TGF-β1 (transforming growth factor, beta) to decrease the cell surface expression of the activating receptor NKG2D (NK cell receptor D) inhibiting NK cell functions [36]. They also demonstrated that the miR-23a in vesicles operates as an additional immunosuppressive factor, since it directly targets the expression of the cytotoxic marker CD107a in NK cells (Table 2). This alteration was also observed with exosomes from MPM by Clayton et al. They showed the capacity of MPM-derived exosomes to reduce the expression of NKG2D by NK cells and CD8^+^ T cells through the transfer of TGF-β1. This downregulation of NKG2D induces the suppression of the NKG2D-dependent killing function of NK cells, as well as the cytotoxic ability of CD8 + T cells [37]. Moreover, the same group showed that exosomes derived from pleural fluid of MPM patients express CD39 and CD73, two receptors that can produce extracellular adenosine and downregulate the function of T cells, thus suggesting a contributory role of MPM-derived exosomes in the negative modulation of T cells [38].

## 6. Thoracic Tumor-Derived Exosomes as Pro-Angiogenic Cues

A mechanism favoring tumor growth and subsequent metastatic events is angiogenesis. Multiple studies suggest a role of lung cancer-derived exosomes in this process. Thus, Hsu et al. reported that miR-23a, significantly upregulated in exosomes from lung cancer under hypoxic conditions, directly suppresses its target PHD1 (prolyl hydroxylase 1) in recipient endothelial cells, leading to the accumulation of the hypoxia-inducible factor-1 α (HIF-1 α) and consequently enhancing angiogenesis (Table 2) [46]. Moreover, they showed that this miRNA also inhibits the tight junction protein ZO-1 (zonula occludens 1 protein), thus increasing vascular permeability and cancer trans-endothelial migration (a key step in metastasis). A study by Liu et al. identified the role of another exosomal miRNA in angiogenesis in lung cancer. They found that exosomal miR-21 secretion by cigarette smoke extract (CSE)-transformed human bronchial epithelial (HBE) cells induces elevated levels of VEGF in recipient normal HBE cells and in human umbilical vein endothelial cells (HUVECs), thereby promoting angiogenesis [48]. Another study reported that upon overexpression of the tissue inhibitor of metalloproteinases-1 (TIMP-1) in lung adenocarcinoma cells, miR-210 accumulates in TEXs (in vitro and in vivo), promoting tube formation activity in recipient HUVECs [49]. More recently, it has been shown that leucine-rich-alpha2-glycoprotein 1 (LRG1), enriched in the exosomes derived from NSCLC tissue and cells, mediates a proangiogenic effect via the activation of the TGF-β pathway [41].

Concerning MPM-derived exosomes, the literature is sparser, but also describes the role of EVs in angiogenesis processes. Greening et al. reported that exosomes from MPM significantly increase HUVEC invasion in the transwell assay and angiogenesis in the tube formation assay compared to vehicle (control)-treated cells [45]. Bioinformatic analyses suggest that some protein cargo, such as glycolytic enzyme G6PD (glucose-6-phosphate dehydrogenase) and ENO1 (enolase1), could be responsible for these effects, in line with previous reports indicating a crucial role of glycolysis in specific tumor endothelial cell subtypes (Table 1) [75].

## 7. Thoracic Cancer-Derived Exosomes in Tumor Invasion and Dissemination

Metastasis, or the capacity of cancer cells to invade other organs, is the leading cause of death from cancer. It is a common phenomenon in lung cancer and MPM. In these processes, Rahman et al. reported that exosomes derived from highly metastatic lung cancer cells and human late-stage lung cancer serum-induced EMT (epithelial-to-mesenchymal transition) in HBE cells through the induction of the vimentin expression. EMT is the process by which epithelial cells dramatically alter their shape and motile behavior as they differentiate into a mesenchymal phenotype, and it is an important process in the initiation of metastasis. Another study showed that all lung cancer-derived exosomes, regardless of the cancer type (EGFR-mutated adenocarcinoma, wild-type adenocarcinoma, squamous cell lung cancer), induced an increase of the matrix metalloproteinase 2 (MMP2) activity in cells exposed to TEXs (Table 1) [40]. Several publications have confirmed that exosomal miRNAs play critical roles in different steps of the metastatic processes. He et al. thus reported that miR-499a-5p was upregulated in highly metastatic lung cancer cell line and their exosomes, promoting cell proliferation, migration and EMT through the regulation of the mTOR pathway, while miR-499a-5p knockdown suppresses them (Table 2) [59]. Kim et al. reported an increase of exosomal miR-23a following the induction of EMT in A549 cells [47]. miR-23a is known to regulate TGF-β-induced EMT by targeting E-cadherin in lung cancer cells [76]. In 2017, Wu et al. found that the expression of miR-96 was positively correlated with high-grade and metastatic lung cancers through its target gene LMO7 (LIM domain 7) [77]. Three other exosomal miRNAs, namely miR-193a-3p, miR-210-3p, and miR-5100, have been involved in the invasion of lung cancer cells through STAT3 signaling-induced EMT [52], whereas some have been described in dissemination, such as miR-660-5p [61] or miR-106b [51].

The “seed and soil” theory of tumor metastasis introduces the concept that a receptive environment is required for the development of tumors in distant sites [78]. When looking up the frequency of metastasis of lung carcinomas, there are some preferential sites, such as bone (34.3%), lung (32.1%), brain (28.4%), adrenals (16.7%), and liver tissues (13.4%) [79]. Regarding MPM, a postmortem study showed extrapleural dissemination of 87.7% in the liver (31.9%), spleen (10.8%), thyroid (6.9%), and brain (3.0%) [80]. Numerous articles indicate the important role of EVs in the formation of these pre-metastatic niches, notably through their ability to disseminate throughout the body and their enrichment in tetraspanins involved in the binding/uptake of exosomes in target cells. Liu et al. thus suggested that the tetraspanin-8 (TSPAN8) levels on serum EVs predict metastasis in NSCLC [81]. Interestingly, integrin profiles of exosomes can also be used to address them to specific organs and promote the development of the pre-metastatic niche [82,83].

## 8. Thoracic Tumor-Derived Exosomes as Relevant Biomarkers

### 8.1. Biomarkers of Diagnosis and Cancer Stages

TEXs contain several tumor-associated proteins that could be used as biomarkers (Table 1 and Table 2), and classically EV levels in the peripheral blood of patients with lung cancer are higher than in healthy controls [84] and are correlated with the tumor stage [85]. Of note, by providing an intuitive term in a clinical setting, Gavard’s team has recently dubbed “vesiclemia” the concentration of plasmatic EVs [86]. Huang et al. showed that 80% of NSCLC exosomes were positive for EGFR, whereas only 2% of exosomes from chronic inflammation of the lung carried this protein [32]. This result was confirmed by Clark et al. using triple SILAC quantitative proteomics to characterize the proteins profile of exosomes derived from two NSCLC cell lines and an immortalized normal HBE cell line [87], and by Sandfeld-Paulsen et al. using protein profiling via EV array [42]. This last group has then evaluated the potential of 49 exosomal proteins as prognosis markers using plasma samples from 276 NSCLC patients. They demonstrated that NY-ESO-1, EGFR, recombinant human placental alkaline phosphatase (PLAP), Alix (apoptosis-linked gene 2-interacting protein X) and EpCAM (epithelial cell adhesion molecule) are markers of poor prognosis with increasing concentration levels. However, NY-ESO-1 was the only marker that maintained a significant impact on inferior survival after multiple testing [88]. They also found that CD151, CD171, and TSPAN8 separated strongly patients with lung cancer of all histological subtypes *versus* patients without cancer. Finally, in urine, Li et al. found high expression levels of the LRG1 in exosomes from NSCLC patients compared to those from control subjects. These results suggest that LRG1 may be a candidate biomarker for noninvasive diagnosis of NSCLC [43].

Among all components of the exosomes, miRNAs are probably the most widely studied biomarkers. The first miRNA signature described in plasma exosomes of NSCLC patients included 12 miRNAs: miR-17-3p, miR-21, miR-106a, miR-146, miR-155, miR-191, miR-192, miR-203, miR-205, miR-210, miR-212, miR-214 [62,89]. Next in 2013, Cazzoli et al. identified a four miRNAs panel (miR-378a, miR-379, miR-139-5p and miR-200b-5p) to discriminate between lung cancer patients and healthy smokers, and a six miRNAs panel (miR-151a-5p, miR-30a-3p, miR-200b-5p, miR-629, miR-100 and miR-154-3p) to discriminate between lung adenocarcinoma and lung granuloma [53]. From Rabinowits and Cazzoli signatures, Lin et al. confirmed the importance of miR-205-5p and miR-200b in lung cancer exosomes in the discrimination of patients with lung cancer or pneumonia [54]. Another study conducted by Rodriguez et al. compared exosomes from the plasma of 30 NSCLC patients and 75 controls to exosomes from bronchoalveolar lavage (BAL) [63]. They profiled 84 miRNAs and specific signatures were identified as follows according to the source of exosomes (plasma or BAL) and pathology (tumor or control): miR-126 and miR-144 in plasma samples (tumor and control); miR-302a and miR-302c in BAL samples (tumor and control); miR-128 in plasma of control individuals only; and miR-143 in tumor BAL only. In addition, miR-122 was the only tumor-specific miRNA found irrespectively of the source (plasma or BAL). In pleural lavage again, Roman-Canal et al. identified 3 miRNAs (miRNA-1-3p, miRNA-144-5p and miRNA-150-5p) that discriminate accurately between control pleural fluids and pleural lavage from lung cancer [64]. More recently, but in serum this time, Zhang et al. showed that miR-17-5p was significantly upregulated in exosomes from NSCLC patients compared with the healthy controls [65]. Interestingly, Jin et al. reported a different panel of 4 miRNAs (let-7b, Let-7e, miR-23a-3p, miR-486) enabling the discrimination of NSCLC patients from healthy controls [66]. They also identified 4 adenocarcinoma-specific exosomal miRNAs (miR-181-5p, 30a-3p, 30e-3p, 361-5p) and three squamous cell carcinoma (SCC)-specific exosomal miRNAs (miR-10b-5, 15b-5p, 320b). For lung adenocarcinoma, Fang et al. showed that miR-505-5p was upregulated in EVs from patients compared to the controls [60]. To conclude here, these panels may be promising candidates for the diagnosis of NSCLC and the distinction between lung adenocarcinoma and SCC, but each study identified its own panels leading to difficult interpretations. These differences may be due to the source of the TEXs (plasma, serum, pleural effusion…), the method of miRNA extraction, but also the year of the study because new miRNAs have been continuously discovered in recent years.

In the case of MPM, Munson et al. showed that the most abundant miRNAs in MPM-derived exosomes are tumor suppressors, in particular miR-16-5p. They hypothesized that MPM tumor cells preferentially secrete tumor suppressor miRNAs such as miR-16-5p into exosomes in order to promote tumorigenesis. In fact, the inhibition of exosomes secretion resulted in a significant killing of cancer cells via the force-feeding of MPM cells by exosomes encompassing miR-16-5p [73]. Another study from Cavalleri et al. found that exosomal miR-103a-3p and miR-30e-3p discriminate MPM patients from non-cancerous individuals exposed to asbestos fibers [74].

### 8.2. Prognostic Biomarkers

Exosomal miRNAs profiling can also be used for prognostic biomarkers in lung cancer. In 2011, Silva et al. studied the expression of 365 exosomal miRNAs from the plasma of 28 NSCLC patients [67]. Five selected miRNAs (Let-7f, miR-20b, miR-30e-3p, miR-223 and miR-301) were validated independently in a second cohort and correlated with pathologic parameters and survival (Table 2). Other miRNAs have been associated with poor survival, such as miR-23b-3p, miR-10b-5p and miR-21-5p [50] and miR-378 [68]. Regarding the pathological parameters, exosomal miR-21 showed a significant association with tumor size and the tumor-node-metastasis (TNM) stage and miR-4257 showed a significant association with histological type, lymphatic invasion, and the TNM stage in lung cancer [69]. Apart from miRNAs, long non-coding RNAs (lncRNAs) included in exosomes, such as lincRNA-p21 [90], have also been associated with NSCLC prognosis.

### 8.3. Biomarkers of Response to Treatment and Resistance

Specific circulating miRNAs are differentially expressed between responders and non-responders and bear potential as predictive biomarkers for treatment (Table 2). Thus, Yuwen al. identified the serum exosomal miR-146a-5p as a potential biomarker predicting the efficacy of cisplatin for NSCLC patients [71]. Cisplatin resistant A549/DDP cells and exosomes expressed lower miR-146a-5p levels than nonresistant A549 cells. Its expression decreases in either NSCLC cell lines and exosomes gradually during resistance acquisition. Qin et al. also reported that exosomes derived from cisplatin-resistant lung cancer cells (A549/DDP) can alter other lung cancer cells’ sensitivity to cisplatin in an exosomal miR-100–5p-dependent manner with mTOR (mechanistic target of rapamycin) as its potential target [55]. Exosomal miR-96 previously described as being involved in lung cancer proliferation and migration seemed to also promote cisplatin resistance in the same study. More recently, Ma et al. showed that exosomes isolated from either cisplatin-treated or cisplatin-resistant NSCLC cells conferred chemoresistance to sensitive A549 cells in an miR-425-3p-dependent manner [56].

A response to the tyrosine kinase inhibitors in the NSCLC context has also been investigated by different groups. Zhang et al. identified an upregulation of miR-214 in gefitinib-resistant lung cancer cells (PC-9GR) and their exosomes compared to gefitinib-sensitive lung cancer cells (PC-9). Moreover, they reported that exosomes derived from gefitinib-resistant PC-9GR cells could transfer resistance to its recipient sensitive PC-9 cells, probably through the transfer of miR-214 by the exosome [58]. Jing et al. showed that miR-21 could be transferred from gefitinib-resistant lung cancer cells (H827R) to gefitinib-sensitive (HCC827) via exosomes and thus induce gefitinib resistance in sensitive lung cancer cells through AKT activation [57]. Another group reported the case of the long non-coding RNA H19. They identified its capacity to induce gefitinib resistance in sensitive lung cancer cells when transferred from gefitinib-resistant lung cancer cells through exosomes [91]. Finally, another study from Giallombardo et al. reported that exosomal expression of miR-221 and miR-222 was associated with a good response to osimertinib in EGFR-mutated NSCLC patients [72].

Concerning PD-1/D-L1 treatment, Shukuya et al. showed that several miRNAs, including miR-199a-3p, miR-200c-3p, miR-21-5p, miR-28-5p and miR-30e-3p, were upregulated in the plasma of non-responders NSCLC patients compared to responders [70].

## 9. Targeting Exosomes and Their miRNAs for Therapy

As previously explained, it is actually well known that cancer-derived exosomes can promote tumor growth, proliferation, and metastasis in various cancer types through different mechanisms such as anti-tumor immune suppression, pro-tumor immune activation, angiogenesis, and EMT via the transfer of a specific protein or miRNA cargo to recipient cells. Because of their abundance in exosomes, their pivotal role in the regulation of gene expression, and the evidence that they can exert functional effects in recipient cells, exosomal miRNAs aroused interest as targets for cancer therapy. The idea is that all the previously described miRNAs that are involved in promoting tumor growth, progression, and dissemination could represent valuable therapeutic targets. Indeed, it could be interesting to use exosomes loaded with therapeutic antagomiRs, complementary to the targeted mature oncogenic miRNAs, and inject them locally or through systemic administration [92]. However, the systemic use of modified exosomes can induce deleterious effects since exosomes can interact with tumor cells but also with all other cells throughout the body, thus altering their functions. Thereby, local injection of these exosomes seems to be the path forward. Another possible therapeutic strategy is to control the miRNA sorting from tumor cells to exosomes. In this way, the release of specific miRNAs could modify the TME. As suggested by Thind et al., it could be interesting to target proteins such as hnRNPA2B1 to control the incorporation of some miRNAs into exosomes [93]. A study of glioblastoma and melanoma cells, and their secreted large EVs, revealed the presence of a 25-nucleotide stem loop-forming sequence in the most EV-enriched mRNAs [94]. A meta-analysis would be useful to unravel whether this so-called “zipcode” sequence is conserved across several cancer entities, including lung tissue. Clinically, by incorporating this zipcode sequence into the 3′ UTR of therapeutic RNAs, one could envision the enrichment of EVs with specific RNAs for their subsequent use as therapeutic carriers. However, up to now, our knowledge about RNA sorting is too poor to envisage this in a clinical setting.

While some proposed to target miRNAs, others mentioned the utility of getting rid of exosomes, either by eliminating them from the circulatory system or by stopping their production and secretion by cells. In this sense, Marleau et al. developed a strategy involving extracorporeal hemofiltration of exosomes from the entire circulatory system using an affinity plasmapheresis platform known as the Aethlon ADAPT™ (adaptive dialysis-like affinity platform technology) system [95], but exosome secretion can also be blocked by drugs such as GW4869, a sphingomyelinase inhibitor. Multiple studies used this drug to reverse the deleterious effects mediated by exosomes in cancer progression and thus reported its benefits [96,97]. However, we have to keep in mind that exosomes are important players of inter-cellular communication in physiological conditions as well.

## 10. Therapeutic Potential of EVs in Lung Cancers

Instead of targeting EVs that may induce pro-tumorigenic effects, another approach would consist in taking advantage of these carriers to deliver various payloads to the tumor. For instance, in contrast to chemotherapy and drugs which act systemically, the use of EVs as carriers may provide higher tumor specificity, improved pharmacokinetics, and thus reduced side-effect toxicity. In the following paragraph, we discuss recent developments and potential approaches that could be used to treat lung cancer. Some of these approaches have progressed to the clinic in regenerative medicine, providing arguments for their clinical use in cancers [98].

The cargos which could be delivered by EVs are diverse and not limited to the following: proteins, nucleic acids (mRNA, miRNA, lncRNA, etc.), genome editing tool (silencing RNA, CRISPR/Cas9), drug compounds, and oncolytic viruses. Seemingly, the type of carrier used can be different with cell-derived EVs, or engineered EVs. Depending on the method employed to load chemotherapeutic compounds into EVs (e.g., co-incubation, electroporation, sonication, extrusion), and also the drug considered, the loading efficiency and EV properties vary tremendously [99,100]. However, when compared to drugs alone, their encapsulation within EVs increases their delivery and cytotoxic effect, as shown for paclitaxel [100] and doxorubicin [99] in vitro on lung cancer cell lines. Of note, overcoming drug efflux-based multidrug resistance in cancer is a formidable unmet problem. As such, it appears that paclitaxel-loaded EVs could sensitize resistant cancer cell lines via bypassing this noxious adaptation in tumors.

As described in a section above, depending on their cell origin, EVs have a certain cell-specific tropism that can be exploited to target particular tissues or organs. Interestingly, in a mouse model of pulmonary metastases, intra-nasal injection of macrophage released EVs loaded with paclitaxel led to a decrease in tumor growth [100]. Surprisingly, using immunohistochemistry, the authors found a near-complete co-localization of stained-EVs with lung metastases, but the other organs were not considered in this study. The EV protein (and perhaps also lipid) signature appears to be key in the cell-type-specific EV delivery. A downside of using cell-derived EVs is the lower yield achieved with standard T flask cell culturing. To cope with this hurdle, diverse methods have been utilized to increase EV yield: bioreactor cultures (e.g., stirred or hollow-fiber devices) [101], vesiculation buffers [102], or cytochalasin B-induced vesicles [103]. A recent study made use of a bioreactor to generate a large-scale production of good manufacturing practice standards and clinical-grade mesenchymal stem cell-derived EVs for human trials in preclinical mouse models of pancreatic cancer [104]. However, presently, no such studies were conducted yet on models of lung cancer.

As a substitute for cell-based EVs, the use of engineered lipid nanoparticles would theoretically circumvent the challenge of yielding enough vesicles for clinical application. On the other hand, as it is relatively difficult to undermine and control surface structures and composition of EVs, the biological function of such engineered EVs should be taken with a grain of salt. For technical considerations regarding the production of engineered lipid nanoparticles, we refer the readers to the following review [105]. Recently, Vazquez-Rios et al. engineered lipid nanoparticles that resemble small EVs for their composition and physicochemical properties, which can be used for drug and nucleic acid loading [106]. The nanoparticles were functionalized with the recombinant human integrin α6β4, related to lung organotropism, and successfully targeted to the lung tumor upon intraperitoneal injection. A similar approach has been used in breast cancer, where doxorubicin-loaded EVs, released from dendritic cells transfected with αv integrin-specific iRGD peptide (CRGDKGPDC) fused to the extra-exosomal N terminus of Lamp2b, an exosome membrane-integral protein, preferentially accumulated into the tumor and reduced tumor growth without inducing cardiac damage [107]. Of note, αv integrin is a prognostic marker for overall survival and disease-free survival in NSCLC [108], suggesting that using the iRGD peptide to deliver EV-payloads could be translated to lung cancers. Compared to cell-derived EV systems, engineered lipid nanoparticles have several advantages for clinical applications: (i) limited preparation time (10 minutes); (ii) higher yield of particle production; and (iii) easier approach than culturing patient-derived cancer cells for autologous EV delivery. However, such *modus operandi* has cons and important questions that remain to be answered. For instance, autologous or allogenic exosomes collected from patients may have immune-privileged status that allows a limited immune response and reduced drug clearance by macrophages. What is the toxicity profile of such nanoparticles? And is the homing message the same on these (relatively simple) engineered lipid nanoparticles? A similar approach using lipid hybridized EVs would partially prevent this limitation, as part of the membrane homing zip code is preserved [109].

Oncolytic virotherapy has emerged as a promising approach to treat cancer [110]. Indeed, the approval of the first oncolytic virus for melanoma treatment (Imlygic) has opened new perspectives to ameliorate cancer treatment with limited curative options (e.g., metastatic lung cancers). However, such virotherapy is still facing a number of hurdles including: (i) clearance of oncolytic viruses; (ii) downregulation of virus-recognizing receptors on tumor cells thereby limiting their entry; (iii) activation of intracellular antiviral defense mechanisms; and (iv) inefficacy in controlling advanced tumor stages [111]. In vitro, A549 lung cancer cells infected by oncolytic adenovirus release large EVs containing functional oncolytic adenovirus [111,112]. Interestingly, such EV-containing oncolytic adenoviruses were efficiently delivered in vivo, regardless of the presence of blocking antiviral antibodies, suggesting a coxsackie-adenovirus receptor (CAR)-independent entry. Moreover, when virus-containing EVs were combined with paclitaxel, their tumor cell killing ability was highly increased [112,113]. Altogether, this suggests that large EVs are capable of facilitating the entry of adenovirus into cancer cells and to augment their oncolytic efficiency. Of note, the cell-killing effect of EV formulations was independent of the origin of the cancer cell line EV used [112], which may indicate a versatile application for oncolytic adenovirus-based EV systems.

## 11. EV-Based Clinical Trials in Lung Cancer

DC-derived exosomes, coined “dexosomes”, were envisioned as possible anticancer vaccinations in NSCLC. Patients underwent leukapheresis to generate dendritic cells from which small EVs were collected and loaded with different formulations of MAGE tumor antigens [114] or with interferon-γ plus MHC class I- and II-restricted tumor antigens [115]. A first phase I study indicated mild adverse effects causally related to the use of dexosomes [114]. Consequently, a phase II clinical trial enrolled 22 patients to test the interferon-loaded dexosomes and was well tolerated, except in one patient where grade 3 hepatotoxicity occurred (NCT01159288). Overall, although EVs could stimulate immune responses and promote anti-tumor responses, results from this trial indicate limited clinical outcomes and no objective tumor response.

Recently, a group has initiated a promising pilot human clinical trial on 11 patients with advanced lung cancer and malignant pleural effusion to assess the safety and feasibility of intrapleural infusion of autologous tumor-derived EVs loaded with methotrexate (TMPs-MTX) [116]. This trial is backed up by solid in vitro and in vivo evidence showing that TMPs-MTX induced higher lung cancer cell apoptosis, preferentially localized into lungs and tumors after intrapleural administration, and lowered the pleural tumor burden in mouses. Additionally, TMP-MTX treatment fostered the percentage of immune effector cells and promoted a tumor-suppressive immune microenvironment. In line with these data, patients treated by autologous TMP-MTX showed an improved objective clinical response rate with a reduction in the tumor, while only mild adverse effects were reported. These promising results are currently being strengthened by the recruitment of additional patients in an opened clinical trial (NCT02657460).

## 12. Conclusions

Cellular interactions within the TME lead to tumor growth and progression and may favor immune evasion. EVs have a crucial role in these interactions through their encapsulated biologically active molecules. Identifying exosomal miRNAs involved in this communication is thus necessary to develop new directed therapeutics (Figure 3).

## Figures and Tables

**Figure 1 ijms-21-06024-f001:**
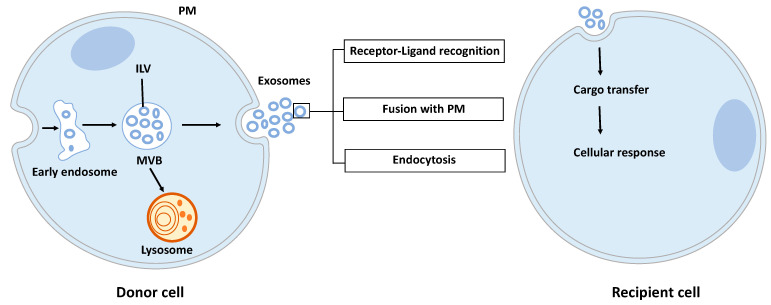
Exosome biogenesis, secretion, and capture. Exosomes are formed by the inward budding of intraluminal vesicles (ILV) during the maturation of the early endosomes into the multivesicular body (MVB). ILV become exosomes after their release into the extracellular space, thus after the fusion of the MVB with the plasma membrane (PM). Released exosomes can interact with the recipient cell through receptor-ligand recognition, fusion with PM, or endocytosis. Cargo can be then transferred into the cytoplasm of the recipient cell where it can exert its effects.

**Figure 2 ijms-21-06024-f002:**
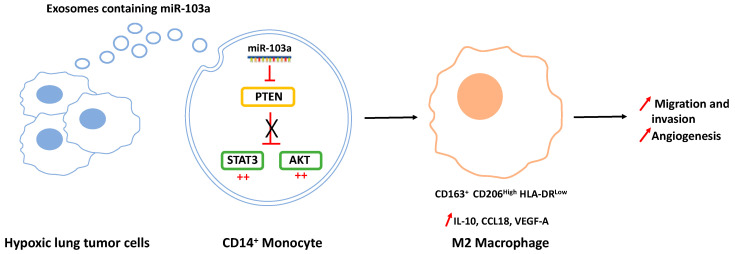
Hypoxic lung cancer-derived exosomes increased M2-type macrophage polarization via miR-103a transfer. Exosomal miR-103a decreases the level of PTEN in CD14^+^ monocytes, allowing the activation of the Pi3K/AKT and STAT3 signaling pathways (denoted by the red ++) and the polarization of macrophages towards the pro-tumoral M2 phenotype with IL-10, CCL18, and VEGF-A secretion, thus promoting cancer migration, invasion, and angiogenesis.

**Figure 3 ijms-21-06024-f003:**
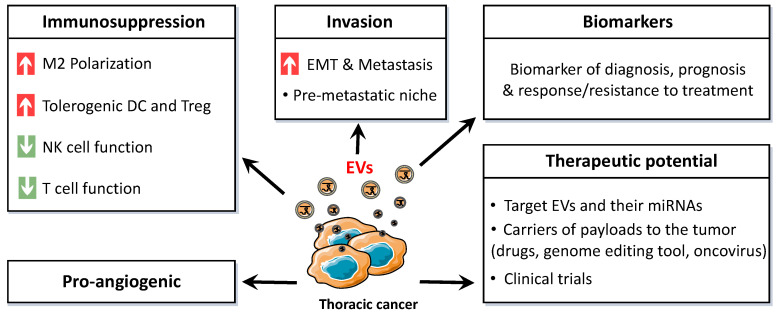
Schematic representation of the broad range of effects and applications of thoracic cancer-derived extracellular vesicles (EVs). M2: type 2 macrophage, DC: dendritic cell, Treg: regulatory T lymphocyte, NK: natural killer, EMT: epithelial-to-mesenchymal transition.

**Table 1 ijms-21-06024-t001:** The role of the exosomal protein cargo in thoracic cancers.

Exosomal Component	Origin of the Exosomes	Role	Cancer	Ref.
**TGF-ß1**	Hypoxic LC cell lines	NKG2D decrease NK inhibition	**LC**	[36]
**Vimentin**	HBE cells	Invasion	**LC**	[39]
**EGFR**	Serum	EMT Cancer invasion Metastasis	**AD**	[40]
**HSP72**	Lung AC cell lines	MDSC promotion Immune suppression	**AD**	[34]
**LRG1**	NSCLC cell lines and tissues	Angiogenesis	**NSCLC**	[41]
**NY-ESO**	Plasma	Prognosis and overall survival	**NSCLC**	[42]
**LRG1**	Urine	Diagnosis	**NSCLC**	[43]
**CD151, CD171, TSPAN8**	Plasma	LC diagnosis following histological subtypes	**NSCLC**	[42]
**EGFR**	LC biopsies	Tolerogenic DCs, Treg Immune suppression	**NSCLC**	[32]
**PD-L1**	LC cells	T cell suppression	**NSCLC**	[35]
**TGF-ß1**	Pleural effusions	NKG2D reduction NK and CD8^+^ T cells suppression	**MPM**	[44]
**CD39, CD73**	Pleural effusions	Adenosine production T cell suppression	**MPM**	[38]
**G6PD, ENO1**		Angiogenesis	**MPM**	[45]

NSCLC: non-small cell lung cancer, LC: lung cancer, AD: adenocarcinoma, SCC: squamous cell carcinoma, CSE-HBE: cigarette smoke extract transformed human bronchial epithelial cells, MPM: malignant pleural mesothelioma, PAE: past asbestos exposure, TGFβ1: transforming growth factor, beta 1, EGFR: epidermal growth factor receptor, HSP72: heat shock 70kDa protein 1A, LRG1: leucine-rich alpha-2-glycoprotein 1, TSPAN8: tetraspanin 8.

**Table 2 ijms-21-06024-t002:** Roles of the exosomal miRNA cargo in thoracic cancers.

Exosomal Component	Origin of the Exosomes	Role	Cancer	Ref.
**miR-23a**	Hypoxic LC cell lines	Targets CD107a NK inhibition	**LC**	[36]
	Hypoxic lung cancer cells	Angiogenesis vascular permeability	**LC**	[46]
	A549 cells	EMT	**LC**	[47]
**miR-21**	CSE-HBE cells	Angiogenesis	**LC**	[48]
**miR-210**	Lung AC cells	Angiogenesis	**LC**	[49]
**miR-23a**	Hypoxic LC cells	Angiogenesis	**LC**	[46]
**miR-96**	Serum	Metastasis	**LC**	[50]
**miR-106b**	Serum	Metastasis	**LC**	[51]
**miR-199a-3p, miR-210-3p, miR-5100**	Plasma	Metastasis	**LC**	[52]
**miR-378a, miR-379, miR-139-5p, miR-200b-5p** **miR-151a-5p, miR-30a-3p, miR-200b-5p, miR-629, miR-100, miR-154-3p**	Plasma	Dividing LC from non-LC Dividing AC from granuloma	**LC**	[53]
**miR-205-5p** **miR-200b**	Pleural effusions	Diagnosis	**LC**	[54]
**miR-100-5p**	A549 cells	Cisplatin resistance	**LC**	[55]
**miR-425-3p**	A549 cells	Cisplatin resistance	**LC**	[56]
**miR-21**	HCC827	Gefitinib resistance	**LC**	[57]
**miR-214**	PC-9 cells	Gefitinib resistance	**LC**	[58]
**miR-499a-5p**	Highly metastatic LC cell lines	EMT, proliferation, migration	**AD**	[59]
**miR-505-5p**	Plasma	Diagnosis	**AD**	[60]
**miR-660-5p**	Plasma	Metastasis	**NSCLC**	[61]
**miR-17-3p, miR-21, miR-106a, miR-146, miR-155, miR-191, miR-192, miR-203, miR-205, miR-210, miR-212, miR-214**	Plasma	Diagnosis	**NSCLC**	[62]
**miR-126, miR-144, miR-302a and miR-302c**	Plasma BAL	Diagnosis	**NSCLC**	[63]
**miR-1-3p, miR-144-5p, miR-150-5p**	BAL	Diagnosis	**NSCLC**	[64]
**miR-17-5p**	Serum	Diagnosis	**NSCLC**	[65]
**Let-7b, Let-7e, miR-23a-3p, miR-486** **miR-181-5p,30a-3p, 30e-3p, 361-5p** **miR-10b-5, 15b-5p, 320b**	Plasma	Diagnosis Diagnosis Diagnosis	**NSCLC**	[66]
**Let-7f, miR-20b, miR-30e-3p, miR-223 and miR-301**	Plasma	Prognosis	**NSCLC**	[67]
**miR-23b-3p, miR-10b-5p and miR-21-5p**	Plasma	Overall survival	**NSCLC**	[50]
**miR-378**	Serum	Overall survival	**NSCLC**	[68]
**miR-21, miR-4257**	Plasma	Prognosis/recurrence	**NSCLC**	[69]
**miR-199-a, miR-200c-3p, miR-21-5p, miR-28-5p, miR-30e-3p**	Plasma	Prediction of PD1/PD-L1 treatment	**NSCLC**	[70]
**miR-146a-5p**	Serum	Prediction of cisplatin response	**NSCLC**	[71]
**miR-221, miR-222**	Plasma	Prediction of osimertinib response	**NSCLC**	[72]
**miR-16-5p**		Diagnosis	**MPM**	[73]
**miR-103a-3p** **miR-30e-3p**	Plasma	Discriminate MPM from non-MPM with PAE	**MPM**	[74]

NSCLC: non-small cell lung cancer, LC: lung cancer, AD: adenocarcinoma, SCC: squamous cell carcinoma, CSE-HBE: cigarette smoke extract transformed human bronchial epithelial cells, MPM: malignant pleural mesothelioma.

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
