# Peer review of "Critical Roles of Tumor Extracellular Vesicles in the Microenvironment of Thoracic Cancers"

_ijms, 2020, doi:10.3390/ijms21176024_

Round 1

Reviewer 1 Report

The paper has solid scientific statements and is a very interesting article for the readers of the Journal

Very good work

Author Response

We thank the reviewer for its very positive comments.

Reviewer 2 Report

Line 78. Make mention of few more online databases (Vesiclepedia, Exocarta and EVpedia).

The only limitation is the absence of comprehensive information about the approaches for production of engineered lipid nanoparticles. The authors could expand the target audience and discuss few more approaches for increasing EVs yield (bioreactor cultures, vesiculation buffer, extrusion through polycarbonate filter, cytochalasin B-induced vesicles): doi:10.1172/jci.insight.99263, doi:10.1021/ac301776j, doi:10.1039/c2ib20022h, doi: 10.1038/s41598-020-67563-9.

Author Response

We thank the reviewer for its very helpful comments. We have now improved our manuscript accordingly.

Line 78. Make mention of few more online databases (Vesiclepedia, Exocarta and EVpedia).

According the reviewer’s comment, we have added the other available online databases and their references.

The only limitation is the absence of comprehensive information about the approaches for production of engineered lipid nanoparticles. The authors could expand the target audience and discuss few more approaches for increasing EVs yield (bioreactor cultures, vesiculation buffer, extrusion through polycarbonate filter, cytochalasin B-induced vesicles): doi:10.1172/jci.insight.99263, doi:10.1021/ac301776j, doi:10.1039/c2ib20022h, doi: 10.1038/s41598-020-67563-9.

We thank the reviewer for this very valuable comment. We have now added few sentences in part 10 about EV productions.

Reviewer 3 Report

Thank you authors for giving me the opportunity to review your work and learn interesting new facts on exosomes in cancer biology. The manuscript is well written and covers most of the literature associated with this work. There are some minor corrections/suggestions that I have which if you can incorporate should make the manuscript even better. These are:

  1. In the second paragraph of the introduction, exosomes should take center stage rather than tumor-associated macrophages. As a suggestion, you should describe exosomes first, them tumor exosomes, then cargo and then extracellular communication. 
  2. As you already have mentioned in section 2.3. that exosomes are loaded with specific molecules then the introductory paragraph to section 2 should be modified accordingly.
  3. Line 393: CRISPR/Cas9 in the nucleic acid part doesn't seem right.
  4. Lines 445-447: can be deleted.
  5. Section titles need to be improved.
  6. Why do the tables have row dividing lines?
  7. An image of the overall conclusion of the manuscript should be added.

Author Response

We thank the reviewer for its very helpful comments. We have now improved our manuscript accordingly.

  1. In the second paragraph of the introduction, exosomes should take center stage rather than tumor-associated macrophages. As a suggestion, you should describe exosomes first, them tumor exosomes, then cargo and then extracellular communication. 

The reviewer is right, we have now removed this part from the introduction and added it at the beginning of part 4.

  1. As you already have mentioned in section 2.3. that exosomes are loaded with specific molecules then the introductory paragraph to section 2 should be modified accordingly.

We modified the document accordingly.

  1. Line 393: CRISPR/Cas9 in the nucleic acid part doesn't seem right.

Many apologies for the confusion. We have modified this sentence which is hopefully clearer now.

  1. Lines 445-447: can be deleted.

OK.

  1. Section titles need to be improved.

We have now improved our titles.

  1. Why do the tables have row dividing lines?

We believed that it was easier to read as follows the table, but obviously not. The reviewer is right, we have now deleted it.

  1. An image of the overall conclusion of the manuscript should be added.

We have now added a concluding image.